# A novel CNN gap layer for growth prediction of palm tree plantlings

T. Ananth Kumar[1], R. Rajmohan[2], Sunday Adeola Ajagbe[3], Tarek Gaber[4,5]*, Xiao-Jun Zeng[6], Fatma Masmoudi[7]

**1** Computer Science and Engineering, IFET College of Engineering, Valavanur, Viluppuram, India,
**2** Department of Computing Technologies, SRM Institute of Science and Technology, Kattankulathur, Tamil Nadu, India, **3** Department of Computer & Industrial Production Engineering, First Technical University Ibadan, Ibadan, Nigeria, **4** Computer Science & Software Engineering, University of Salford, Manchester, United Kingdom, **5** Faculty of Computers and Informatics, Suez Canal University, Ismailia, Egypt, **6** Department of Computer Science, University of Manchester, Manchester, United Kingdom, **7** College of Computer Engineering and Sciences, Prince Sattam Bin Abdulaziz University, Alkharj, Saudi Arabia

\* t.m.a.gaber@salford.ac.uk

## Abstract

Monitoring palm tree seedlings and plantlings presents a formidable challenge because of the microscopic size of these organisms and the absence of distinguishing morphological characteristics. There is a demand for technical approaches that can provide restoration specialists with palm tree seedling monitoring systems that are high-resolution, quick, and environmentally friendly. It is possible that counting plantlings and identifying them down to the genus level will be an extremely time-consuming and challenging task. It has been demonstrated that convolutional neural networks, or CNNs, are effective in many aspects of image recognition; however, the performance of CNNs differs depending on the application. The performance of the existing CNN-based models for monitoring and predicting plantlings growth could be further improved. To achieve this, a novel Gap Layer modified CNN architecture (GL-CNN) has been proposed with an IoT effective monitoring system and UAV technology. The UAV is employed for capturing plantlings images and the IoT model is utilized for obtaining the ground truth information of the plantlings health. The proposed model is trained to predict the successful and poor seedling growth for a given set of palm tree plantling images. The proposed GL-CNN architecture is novel in terms of defined convolution layers and the gap layer designed for output classification. There are two 64×3 conv layers, two 128×3 conv layers, two 256×3 conv layers and one 512×3 conv layer for processing of input image. The output obtained from the gap layer is modulated using the ReLU classifier for determining the seedling classification. To evaluate the proposed system, a new dataset of palm tree plantlings was collected in real time using UAV technology. This dataset consists of images of palm tree plantlings. The evaluation results showed that the proposed GL-CNN model performed better than the existing CNN architectures with an average accuracy of 95.96%.

**Data Availability Statement:** The utilized dataset is new and made publicly available in the Kaggle repository at https://www.kaggle.com/datasets/rajmohan89/palm-tree-plantlings-health-prediction

Also the code is available here https://github.com/tananthkumar/plantlingsgrowth.

**Funding:** This study is supported via funding from Prince Sattam bin Abdulaziz University project number (PSAU/2023/R/1444). The funders had no role in study design, data collection and analysis, decision to publish, or preparation of the manuscript.

**Competing interests:** The authors have declared that no competing interests exist.

# 1 Introduction

A seedling is a plant grown from the grain it came from. The embryo inside the seed is made up of a small root and a shoot simultaneously [1]. Before any other part of the plant grows from the seed, the root grows. As the plant grows, its roots start to pull water from the soil. This makes the plant stick to the ground. The shoot grows from the seed in the end [2]. After that, the root will start to take water into itself. The palm tree's benefit expanded when growers recognized the tree's salt and drought tolerance, as well as its role in fighting desertification [3]. The tree has the potential to reduce both the temperature of the atmosphere and the level of pollutants produced by industrial activity. The palm tree's health prediction system with symmetrical design has introduced a new aspect to its consequences for future environmental betterment [4]. For smart farming, the users need a reliable source of information about how things are going now. So, automated monitoring and growth interpretation of palm tree seedlings gives farmers a new way to manage their resources that is based on technology instead of the old way they did it in the past. The monitoring and growth prediction method also explains how plants grow and how healthy they are. This is especially helpful for tracking how old plants are and how often they die so that they can be used to grow palm trees in the future [5]. It is important to obtain reliable detection for palm trees in huge regions because of the economic gain and detrimental environmental consequences. Such precise observations would enhance plantation planning, oil palm productivity, and decrease personnel and fertilizer use by allowing for more precise monitoring of the plantations' growth and more intelligent control of the farms' operations [6].

Technology like the Internet of Things (IoT) and Unmanned Aerial Vehicles (UAV) could have a significant effect on agronomic crops and plants [7, 8]. Due to the common occurrence of plant illness, manual disease detection is a critical task in the agricultural sector. The failure to take the necessary precautions, however, has real-world consequences for plants, diminishing crop quality, quantity, and efficiency. Finding a way to automatically detect plant infections is helpful since it cuts down on the amount of manual inspection work required for large-scale production farms.

ML has been widely used to evaluate large quantities of agricultural data, such as crop type prognosis from Landsat images, agricultural production, irrigation requirements, insect, and epidemic attacks, and weed recognition [9]. This method is used to increase harvests of corn, grapes, soybeans, wheat, chili peppers, paddy, and, most notably, palm oil by automating several tasks, including tree counting and crop growth assessment [10]. Numerous research on useful resources and ML methods for the palm oil sector was undertaken. Research has investigated the use of remote sensing, breeding, and technology to keep tabs on palm oil farms [11]. Bioenergy manufacturing technologies for dealing with fruits and palm oil waste have been evaluated in another research [12]. Researchers [13, 14] explored the use of ML to identify proximal image-based nutritional deficiencies in palm oil trees. Predicting agricultural yields, such as palm oil, has been studied extensively, and so has the application of ML features in automated fruit grading via image processing [15]. However, most of the existing research did not perform a comprehensive literature search.

Deep learning is a subset of machine learning that has grabbed the interest of working in agricultural sectors. Deep learning has recently been a prevalent approach to overcoming issues in computer vision, natural language processing, and video processing [16]. The potential to comprehend complex data is a substantial advantage of deep learning. Conventional statistics and machine-learning approaches can have substantial issues when attempting to extract usable information from data, but deep learning mitigates this necessity. One potential area of study is the use of Deep Learning algorithms for the automated extraction of high-

abstraction data representations (features). These algorithms build a hierarchical structure for learning and representing data, with each layer defining the next one up in terms of the features it learned from the previous one [17].

Daily, farms generate millions of data sets on various topics, including heat, sediment, water consumption, weather conditions, and more. This data is utilized in real-time with the assistance of artificial intelligence and deep learning models to gain insights such as the optimal time to sow seeds, crop selection, hybrid seed selection for increased yields, and several other similar topics [18]. The term "precision agriculture" refers to accurate and reliable harvesting; this has been made possible by using deep learning systems. With the assistance of DL technology, problems such as nutrient deficiencies, plant diseases, and pest infestations can all be identified. Using DL methods, we can pinpoint the exact location of weeds and determine the most effective herbicide to use in that specific area. By doing this, we save revenue and lessen our reliance on herbicides [19]. The growth of crops is a critical feature in the agricultural yields of farming. In practice, some plantlings never arise from the grave or develop properly. This will inevitably lead to a reduction in productivity [20]. Seed germination monitoring seems to be a difficult task.

Additionally, deep learning has been implemented to crop cultivation to cut production costs and hence increase agricultural productivity. Oil palm plants were classified using a sliding window algorithm using a high-resolution satellite photo [21]. They trained and improved the convolutional neural networks (CNN) system using data from a manual count. Then, using the feature extraction technique [22], all observations were estimated on images. Their study's findings indicated the ability to discriminate between damaged and healthy plants [23]. Deep learning can be used in a number of tried-and-true ways, such as with recurrent neural networks (RNNs), long short-term memory networks (LSTMs), and CNN among others. Even though RNNs and LSTMs have a lot in common and are often used to analyse and predict time series problems, RNNs can be taught to do tasks that require long-term memory. Even though RNNs and LSTMs share a lot, this is the case. On the other hand, the most common type of deep neural network used for computer vision and finding objects is the CNN [24].

From the literature, it was found that there are many ML studies [10–14] and DL studies [39–43] have been proposed for finding and counting oil palm trees, to estimate yields, monitor crops, and identify nutritional deficiencies etc. However, comprehensive (monitoring growth and counting the healthy and unhealthy plantlings of palm tree with high accuracy) study on palm tree plantlings is yet to be researched. In this study, a novel Gap Layer modified CNN architecture (GL-CNN) model is proposed to streamline palm tree seedling growth prediction with high accuracy. The images of the plantlings are captured using UAV drone technology. The ground truth of plantlings' health is obtained using IoT technology. The plantlings are tracked during their growth using a drone, and the temperature and humidity are assessed using a grove sensor (DHT11) which is connected to the GND of the raspberry pi. Their progress is observed in plantlings to predict healthy growth.

The major contributions of the research paper are as follows.

- A novel architecture (GL-CNN) has been proposed for the CNN in terms of defined conv layers and the gap layer. There are two 64×3 conv layers, two 128×3 conv layers, two 256×3 conv layers, and one 5123 conv layer for processing the 128×3 input image of palm tree seedling. The output layer receives the processed information from the conv layers and fine tunes them. To determine the classification of plantlings, the output from the gap layer is modified using the SoftMax classifier.

- The novel GL-CNN architecture using drone imagery has been proposed for health and growth prediction for palm tree plantlings.

- A new dataset of palm plantlings has been collected and used to train and evaluate the proposed GL-CNN architecture. The light conditions have been considered while collecting this dataset and drone imagery technology has been used. A collection of 257 images of individual oil palm tree plantlings classified as healthy, or sick were used as training data. In-depth analysis and comparison of existing studies, feature set evaluation, and critical analysis of ML-based palm oil prediction systems. It is shown that modern deep neural networks are a much more accurate approach for predicting plantlings growth than the conventional machine learning approaches, and specified designed CNNs such as the GL-CNN proposed in this paper can achieve a very high accuracy.

The layout of the paper is organized as follows. The following section offers an analysis of the associated literature. The third section offers a holistic clarification of the proposed work's technique. Section 4 gives experiments of the program's design. Section 5 outlines the research outcome and discusses the proposed framework. Section 6 defines the results and their comparison with previous methods. Lastly, in the final phase of the research, there is a summary and guidelines for some further investigation.

## 2 Literature review

Monitoring is essential to restore the environment, but it can be hard to do on a large scale, especially when plants are just getting started. Seeds and seedlings are especially vulnerable parts of a plant's life cycle, but they are tiny and do not have any distinguishing features. This makes it hard to keep track of them [25]. This makes it hard to keep an eye on seeds and young plants. There is a need for technical approaches that can give people who do restoration work plant-based monitoring systems that are high-resolution, quick, and scalable in the past, keeping track of seedlings required expensive field surveys where they had to be identified, and counted while still on the ground [26]. Machine learning and deep learning architectures are two of the newest technologies that offer improvements in ecological monitoring. They could help plants do better in many ways. Capturing the images is a tedious task in disease classification. Many notable works have been done. The researchers in [27–29] simulated a 3D environment and deployed a virtual pinhole camera anywhere in the three-dimensional space surrounding the internal logistics system. In addition, they used multi-view geometry among virtual cameras to get a better look at the trajectories. As a result, they were able to conduct many well-controlled experiments and collect a massive dataset of thrown object trajectories, which we used to successfully train a bidirectional long short-term memory (LSTM) deep neural network. Successful real-time trajectory prediction using a trained neural network has been achieved.

### 2.1 Machine learning for plantlings monitoring

The researchers of [30] applied random forest, an artificial neural strategy, to discern between healthy and damaged plants, and Contour of Focused Gradients was used to gather picture characteristics, and their method scored a recognition rate of 92. They deployed ML algorithms and computer vision technologies to evaluate and diagnose early vegetative infections in this report [31]. In their study [32], they collected a plant leaf image and evaluated it to assess the plant's overall health. They deployed SVM and ANN approaches to spot phytopathogens. SVM, backpropagation, and region of interest are the technologies utilized in these publications. SVM worked well as an outcome of these experiments, including image processing processes. They implemented K-means clustering for image extraction and SVM for the categorization of paddy leaf spots. They acquired a trained accuracy of data of 92% and testing

completeness of data of 74 basis points. Linear interpolation and characterization concepts are presented in this work [33]. The main objective of these methodologies is to forecast or effectively detect input information by using instances. The authors of [34] devised an uncomplicated and straightforward system for evaluating good and diseased tomato leaves. The database included 200 pictures captured with a camcorder. Although the performance gained with the supervised learning approach was satisfied, the decision tree has some limitations—for example, if noisy data is employed, overfitting might arise. The authors of [35] applied random forest, an artificial neural algorithm, to discern between healthy and sick leaves, and Histogram of Oriented Gradient (HOG) to retrieve image features, and their approach attained 92% of overall accuracy. This paper [36] presented an evaluation of the effect of distinct procedures that are simulated to foretell the damaged proliferation of a lettuce sprout. On the testing data, the highest-rated functioning network-Alexnet seems to have a high value of 94%, with a minimum error of 0.17. Using machine learning techniques, the researchers were able to analyze the seedling monitoring successfully but the accuracy rate of prediction is much lower. Moreover, no work is done on palm tree seedling monitoring using machine learning techniques. In [37] research, a revolutionary AI system is suggested for categorizing different kinds of fruits. To begin, they used a 2D fractional Fourier entropy grid based on rotation angles to extract information from pictures of fruit. The data was then classified using a five-layer stacked sparse autoencoder. With a dataset containing 18 types of fruit, the proposed technique obtained a micro-averaged F1 score of 95.08% across 10 separate runs.

## 2.2 Deep learning based plantlings monitoring

Deep learning architectures have demonstrated exceptional performance in forecasting plant health monitoring and classification of diseases. In [38] a framework was suggested to identify and classify the disease based on the expulsion Chroma proportion of the in-undated location of paddy plant via wavelet transform, and then a Naïve Bayesian predictor was adhered to eventually label the ailments into three disease labels named leaf spots menace, grain detonation, and brown spot, with an exactness of 89 basis points. This work conducted a detailed evaluation of the diagnosis of diseases in apples and leaves of tomatoes applying CNN approaches. The system was built on a green leaf image dataset containing more than 3000 pictures and obtained an average accuracy of 87%.

Changing climate has posed a danger to food land; severe temperatures and humidity, as well as other osmotic stress, all add value to the evolutionary biology of ailment and herbicides on crops. In [39], authors employed an image processing technique to identify plant illness. They used a dataset of nearly 500 pictures of the healthy and unhealthy plants in which a convolution neural network is used to detect; a semi-supervised method and a supervised algorithm are utilized. In [40], Wani, J.A. et al. used a pre-trained Inception-v3 system, nine major Convolutional networks were possible to perceive pathologies in foliar shots spanning a few groups and determine vegetation and developmental periods of infestations by tracking the number of leaflets of some diverse species diversity. There might not be a great deal of research on cabbage prognosis. If an Inception Network is redesigned, caution is advised to verify that the computational improvements are not squandered. As a result of the uncertainties about the new network's efficacy, customizing Inception architecture for numerous use cases has become a challenge.

The authors in [41] used a deep CNN architecture, to classify the fruit leaf-disease association. They created a dataset from a real-world setting, consisting of 14,181 photos with 10 different labels. Datasets in colour, monochrome, and grayscale are used to test various hypotheses. The AlexNet and SqueezeNet convolutional neural network models were used to train these datasets, with the identical hyper-parameter settings. The results of the trials show

that both models achieve nearly identical identification accuracies of 86.8% and 86.6% on colour photos, respectively, indicating that colour images are useful for categorization.

The article [42] describes ways for classifying plant plantlings using a collection of 4,275 photos of roughly 960 distinct plant families to 12 species at various developmental stages. They implemented an image classification method using a standard Convolutional Neural Network (CNN), and the system attained an accuracy of roughly 93%. The researchers in [43] suggested a novel Faster RCNN framework-based automatic identification approach for hydroponic lettuce plantlings. Their approach employed the High-Resolution Network (HRNet) as the backhaul for extracting features, resulting in dependable and high—dimension expressions. They classified healthier plantlings from unhealthy plantlings with a 94% accuracy. The researchers in [44] used a CNN architecture accompanied by an LSTM-based deep neural system to identify seedling growth. This paper presents a comprehensive visual synthesis and machine learning pathway for classifying three distinct phases of seed germination. With a multiclass labelling dataset, the CNN-based LSTM model achieved 90% accuracy.

According to the aforementioned studies, the overwhelming amount of palm tree seed mapping studies employ conventional data mining algorithms, such as principal component analysis, supervised classification, spectroscopic measures and binarization analysis, blended per-pixel classification, probabilistic reasoning, and DT principle-based entity image recognition. Moreover, the conventional machine learning approaches are less accurate but have wider applications, whereas deep learning is more accurate but has not reached to the same level of adoption like ML ones. Also, there are ramifications and an increased computational cost if CNN is used to forecast palm tree seed development. Furthermore, no studies have been dedicated to using deep learning techniques for prediction of palm tree seedling growth. Predicting the production of tree crops like oil palms is extremely difficult. To comprehend and lessen the effects of these dangers, we need extensive and multifaceted data sets. Typical approaches can't be used to draw a diagram of their interconnections because they deal with unpredictable and dependent parts. Critical restrictions of productivity at the tree and field sectors can be better understood with the help of modern analytics, which must be combined with the very heterogeneous datasets. Therefore, the work in this paper develops a novel GL-CNN architecture along with UAV and IoT technology for palm tree seedling monitoring.

## 3 Proposed GL-CNN model

In bioremediation, the germination rate, seedling sprouting, and initial establishment stages is responsible for 90% seed death rate, which are regarded as the most critical bottleneck [45]. Conventional assessments of seedling growth can be hard in locations where the scope of seed-based ecological restoration efforts has greatly expanded. In various seedling ecosystems, classifying and predicting seedling health to the specific level can be laborious and time-consuming [46]. Consequently, there is a growing demand for technological approaches that provide restorative practitioners with high-resolution, quick, and affordable seedling health monitoring solutions. However, recent advancements in software platform, wireless sensors, and computer vision may allow for significant time and expense reductions in plant seedling monitoring. Many agronomic applications make use of wireless sensor networks, such as remote monitoring of environmental and soil conditions for estimating crop viability. Using environmental parameters such as temperature, moisture, temperature, relative humidity, salinization, and soil conductance as inputs, WSN-based applications can calculate calculates an anticipated irrigation schedule for agricultural areas [47].

Manually detecting crop diseases and pests, using statistical calculations to forecast the amount and estimate the production and loss of crops. These were time-consuming processes

and prone to human error [48]. With the use of data analytics and machine learning, we can extract the most relevant insights from agricultural data and improve precision in farming. Vector models, neural network, regression techniques, fuzzy logic, RNN and CNN are some of the most popular and relevant machine learning techniques used in weather-based crop monitoring [49].

Digital mounted cameras on lightweight unmanned aircraft systems can capture images of the plantlings. Object-detection algorithms like CNN can then be applied to the acquired imagery to assess the growth of plant plantlings. But still the attempt of palm tree seedling monitoring is not yet done in any research. Hence in this study, we design a novel GL-CNN architecture for efficient monitoring and prediction of palm tree seedling health with UAV for image collection and IoT technology for ground truth estimation as shown in Fig 1.

Monitoring is an essential component of efforts to restore the environment, but doing so on a large scale can be difficult, especially in the early stages when plants are still establishing themselves. Even though they are minuscule and devoid of distinguishing characteristics, seeds and seedlings are the most vulnerable stages of a plant's life cycle [34]. Nonetheless, these phases are crucial to the plant's development. This makes it challenging to keep track of them. This makes it challenging to keep a close eye on the seeds and young plants. Restoration professionals require technical approaches that can provide them with high-resolution, rapid, and scalable plant-based monitoring systems [21]. Conventional approaches to plantlings' health tracking have made use of remote sensing tools like proximity sensing devices, spectrometry, machine vision systems, mapping techniques, and drones. There are still some partial difficulties with these technologies that prevent them from being widely used for tailored and long-term plantlings monitoring [17]. These methods are not suitable for precise monitoring of

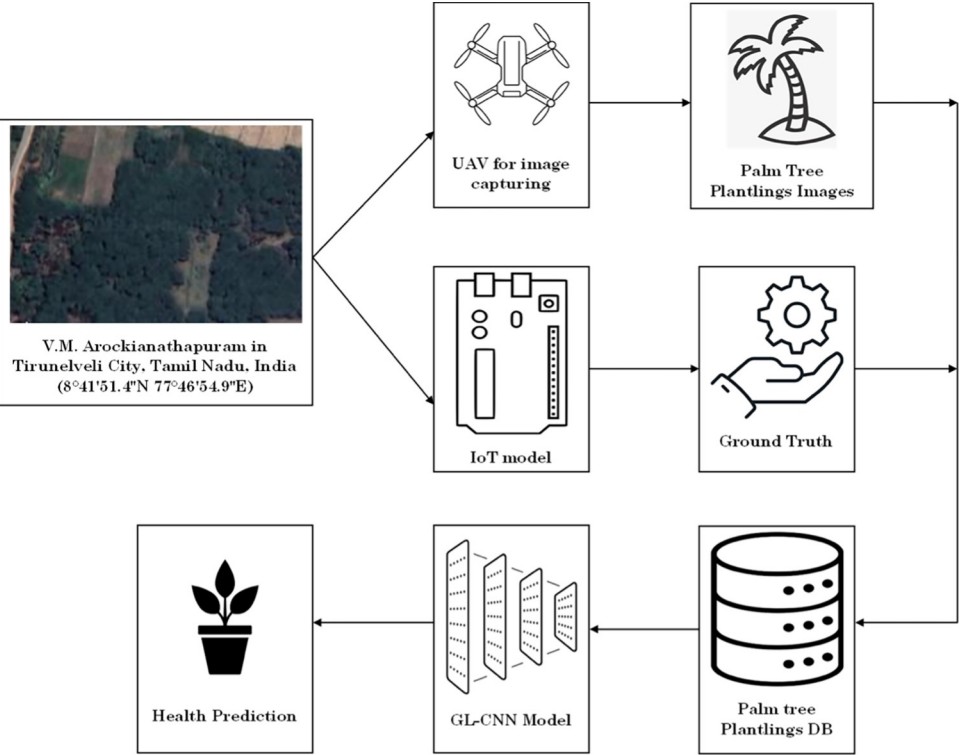

**Fig 1. Schematic workflow of proposed GL-CNN-based model.**

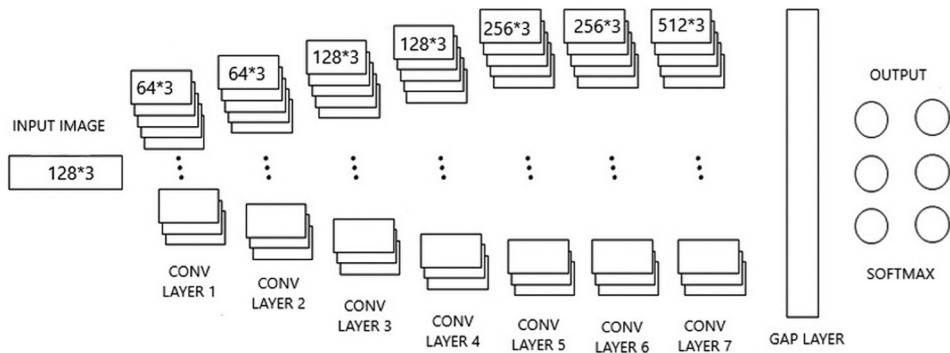

**Fig 2. Proposed GL-CNN architecture.**

plant growth, or the prolonged tracking of the plant's growth. They do not have high spatial or temporal resolution and they can only perform discontinued measurements. In the past, tracking seedlings required conducting costly and time-consuming field surveys during which seedlings had to be located, named, and counted while still in the soil [10]. Machine learning and deep learning architectures are the newest technologies that improve ecological monitoring. To accomplish this, Fig 2 shows that the relatively more minor dimension is stretched to 256, and then the following image is filtered to generate 128*128 images. The filters used are 3x3 kernel size with 64, 128, 256,512 filters at each convolution layer inside every stage.

Table 1 denotes the layered structure schematics implemented in our proposed GL-CNN architecture. It comprises of the components and configuration of our proposed architecture.

**Table 1. GL-CNN configurations.**

| COMPONENTS | CONFIGURATION |
| --- | --- |
| IMAGE INPUT(DATA) | 128*128*3 |
| CONVOLUTION(conv1) | 64*64*3 |
| ReLU (relu1) | ReLU |
| CONVOLUTION(conv2) | 64*64*3 |
| ReLU (relu2) | ReLU |
| MAX POOLING(POOL1) | Max pooling2D, Dropout = 0.25 |
| CONVOLUTION(conv3) | 128*128*3 |
| ReLU (relu3) | ReLU |
| CONVOLUTION(conv4) | 128*128*3 |
| ReLU (relu4) | ReLU |
| MAX POOLING(POOL2) | Max pooling2D, Dropout = 0.25 |
| CONVOLUTION(conv5) | 256*256*3 |
| ReLU (relu5) | ReLU |
| CONVOLUTION(conv6) | 256*256*3 |
| ReLU (relu6) | ReLU |
| MAX POOLING(POOL3) | Max pooling2D, Dropout = 0.25 |
| CONVOLUTION(conv7) | 512*512*3 |
| ReLU (relu7) | ReLU |
| GLOBAL AVERAGE POOLING2D | Global Average pooling layer |
| SOFTMAX | SOFTMAX(CLASSIFIER) |
| LOSS | CATEGORICAL_CROSSENTROPY |
| OPTIMIZER | ADAM |

To retrieve relevant attributes from images, several Convolutional Kernels are utilized. The convolutional layer is a core part of the deep neural network. The gradient attributes are comprised of a set of trainable particles that have a modest dynamic range but stretch across the whole depth of the embedding layer. A single convolutional layer often has several kernels of the same size. In contrast, the Convolutional Layer begins with 64 kernels of various sizes, with the aspect ratio of the kernels typically being the same, and the profundity is almost the same as the multitude of channels [50].

After the first two Convolutional layers have been added, the Overlapping Max Pooling layer is added. After the third and fourth convolutional layers, there are layers of max pooling that overlap, and the fifth convolutional layer is directly connected to the layer before it. After the sixth convolutional layer, there is an Overlapping Max Pooling layer right after it. The final layer in the implementation is the seventh and last convolutional layer. Consequently, a series of one global average pooling is constructed. The GAP layers [51] output is projected into a classifier called SoftMax. Bayes optimization tests [52] also confirmed that ReLU and dropout have collaboration, which implies that using them together is optimal. To reduce the dropout and overfitting problems on a global scale, Bayesian optimization (BO) is a statistical optimization method. BO minimizes or maximizes the objective function using Bayes' Theorem to guide the search. When implemented for hyperparameter tuning, the ReLU objective function becomes costly to evaluate. This can be eradicated by realizing the Bayes technique for parameter tuning in ReLU. CNNs' breakthrough is characterized by their potential to learn detailed mid-level vision descriptions instead of low-level palm tree plantlings parameters, which are prevalent in traditional image classifiers.

Convolution actions utilize 7-dimensional convolution layers and adaptable kernels or filters, with each kernel possessing an additional trainable partiality. The kernels are dragged across the input in "strides" throughout these convolution actions" [53]. In general, the larger the stride, the more the space kernels skip among each iteration. As a result, there were considerably fewer convolutions, and the result size was halved. For each deployment of a specific kernel, a multiplying operation is applied in between the input section and the kernel, with the bias appended to the resultant. It generates extracted features with the convolution layer result. Usually, features were channeled through a kernel function that provides data for the subsequent layer. The below Eq (1) is used to compute the output size of the feature map.

$$Output_{Size} = \frac{(N - F + 2P)}{(S + 1)} \tag{1}$$

where N denotes input size, F denotes kernel size, P denotes padding and S denotes stride.

## 3.1 GL-CNN processing layers

**3.1.1 Overlapping-Max pooling 2D.** Overlapping Max Pool layers are analogous to Max Pool layers because perhaps the progressive windows for which the maxima are estimated to overlap. We proposed an overlapping batch normalization of size 3*3 having strides 2 in this research. Accumulating frames of 3*3 were adopted; with a factor of 2 inter panels. Being contrast to pre-pooling strips of size 2*2 with a stride of 2 yielding the same outcome parameters. The continuous pooling feature leads to a 0.4 percent decline in the top-1 failure rate and a 0.3 percent reduction in the top-5 prediction error [54].

**3.1.2 Normalization layer.** Batch normalization is a recently developed approach that quantifies a mean value from the distribution of the cumulative signal to a neuron across a mini batch of the training phase, which would be used to equalize the summing inputs to those neurons on every set of data [55].

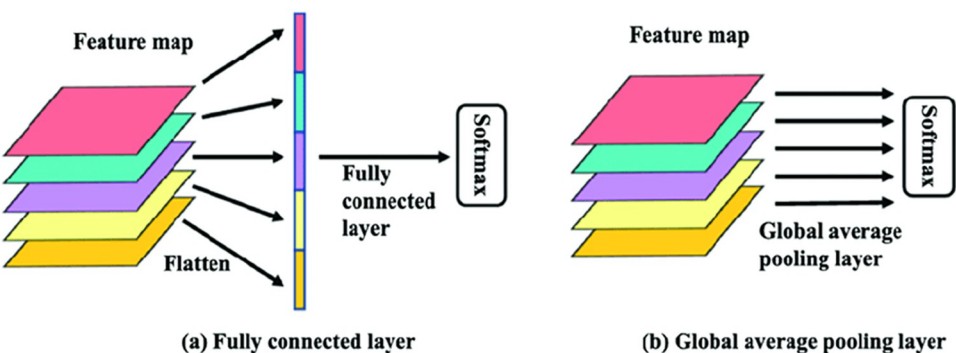

**Fig 3. GAP layer vs. FC layer.**

**3.1.3 Global average pooling layer.** Global Average Pooling is a pooling procedure that is intended to replace ultimately linked layers in traditional CNNs. The working model of GAP is shown in Fig 3. The 2D Global average pooling block accepts a matrix of dimension (input length) x (input altitude) x (input channels) and computes the average of all values across the entire (input size and shape) x (input length) matrix for each of the input channels. Rather than superimposing ultimately linked layers on top of the wavelet coefficients, we aggregate one by one and send the obtained vector right into the SoftMax layer [56].

Global average pooling has virtue over fully connected layers (FC) layers in that it promotes correspondence amongst feature maps and subcategories, making it more ideal for the convolution layout. As a corollary, the extracted features may be simply under-stood as category optimism maps. Another perk of global average pooling is that there are no factors to tweak, thus, fitting during this stratum is minimal. Global average pooling aggregates geometric information, making it even more susceptible to source spatial changes. In conventional CNNs, fully connected layers are typically replaced with a pooling technique called global average pooling. The idea of the final conv layer is to build a feature map for each class that has been established through the classification process. To avoid the need for additional fully linked layers on top of the feature maps, we simply average the maps and feed the resulting vector into the softmax layer.

In addition to alleviating the overfitting issue, pooling enables the learning of invariant features and serves as a regularizer. Equally as crucial is the fact that pooling approaches drastically cut down on both the computing cost and training time of networks.

**3.1.4 Softmax.** SoftMax is advantageous, and it is utilized in CNN for multi-classification. When the SoftMax function is being used in a multi-classification network, it delivers the likelihood for every class, also with the target class, which holds the most significant possibility. Since it turns the output of the neural network's last layer into an effective probability distribution, it is often used to standardize the output of neural networks, which falls around zero and one. It indicates the network output's certain "probability" [57]. The soft max function σ(Z) is defined by Eq 2 which is given below.

$$\sigma\left(Z_j\right) = \frac{e^{Z_j}}{\sum_{k=1}^{N} e^{Z_k}} \qquad (2)$$

The palm tree plantlings growth prediction algorithm is presented below. The input to the algorithm would be preprocessed images and the output will be in the form of values for health and unhealthy growth. The ground truth value obtained through IoT module is compared with predicted result for analyzing the efficacy of the proposed model.

```
Algorithm 1: Palm tree plantlings growth prediction algorithm
Input: Palm tree seedling dataset (Aceraceae) images; Training and
testing ratio, stopping criteria;
Output: Palm tree seedling dataset (Aceraceae) growth prediction, Pos-
itive and Negative polarities, F1-score, Accuracy rate, Precision rate
and Recall rate.
1. Dataset = Palm tree seedling dataset (Aceraceae) images
2. Train = Train GL-CNN model with the palm tree seedling (Aceraceae)
images dataset
3. GL-CNN model = Developed 7 convolutional layers, 3 Max pooling, 1
global average pooling 2D, 1 SoftMax
4. Analyze network (GL-CNN model)
5. Training Option:
   Optimization Algorithm = rmsprop
   Initial Learning Rate = 0.00001
   Max Epochs = 250
   Mini Batch Size = 3*3
6. Load Dataset
7. Count each label in Dataset
8. Resize Dataset to [128*128*3]
9. Convolutional layer [64*64*3]
10. Max pooling2D, Dropout = 0.25
    11.    Split Dataset into [Training Data (80%), Testing Data (20%)]
12. Load GL-CNN model
    13.    Train GL-CNN model /*train the network with the Training
Option and Training Data */
14. Test Trained GL-CNN model /* using Testing Data */
15. Return Test Results
    16.    Test Results = Accuracy rate, Precision rate, Recall rate,
and F1-score
17. If Test Results = Satisfactory, then
    (1) Save the Trained GL-CNN model and Test Results /* for transfer
learning purpose */
    (2) End
18. Else
19. Adjust the Training Option through learning rate
20. Repeat the process until stopping criteria satisfied
```

## 3.2 Structural features of GL-CNN

**3.2.1. ReLU non-linearity.** GL-CNN proves, leveraging Rectified linear units which are a nonlinearity function, the deep CNNs can indeed be developed much faster with the help of saturating receptive fields like Tanh or Sigmoid [58]. The ReLU function is denoted in Eq 3.

$$F(x) = (0, x) \tag{3}$$

The plots of the two functions–tanh and ReLU–are depicted in Fig 4. The tanh function inundates very elevated z values. The slope of the function approaches 0 in specific locations. As a result, gradient descent may be slowed. However, for larger positive values of z, the gradient of the ReLU curve is not close to 0.

This allows the optimization to converge more quickly. The slope remains 0 for negative z values; however, most neurons in a neural network normally have positive values. For the same reason, ReLU wins over the sigmoid function [59]. This kind of architecture has millions of parameters; a major issue arises in terms of over fitting. The major methods to reduce over-fitting are Data Augmentation and Dropout.

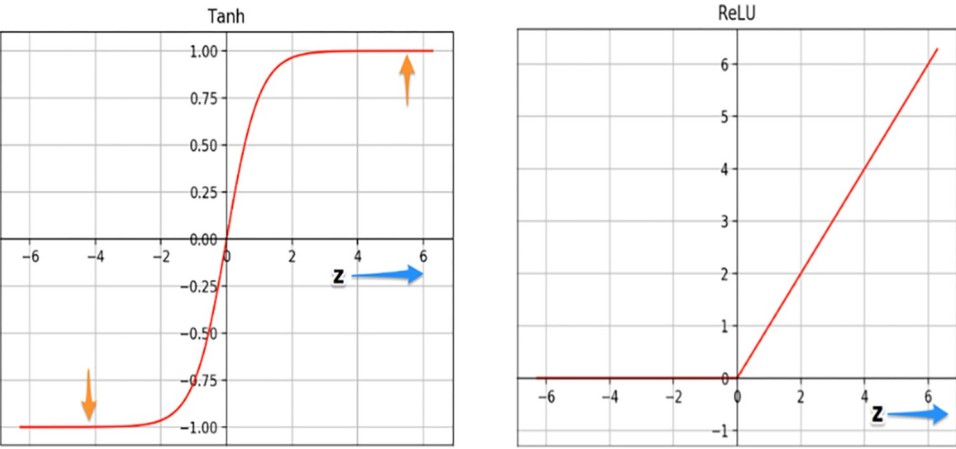

**Fig 4. Activation analysis of tanh and ReLU.**

**3.2.2. Data augmentation.** A regularization approach is data augmentation (a way to prevent overfitting). First, it employs random cropping of input images, as well as rotations and flips, to train the network. Data augmentation is an approach to enhance the quantity of data by adding significantly changed replicas of previously existing data or created from previously existing data [60]. When retraining a machine learning model, it serves as a regularizer and helps to minimize the computational burden.

Fig 5 depicts the randomly cropped images which appear to be highly similar, yet they are not precisely the same. This informs the Neural Network that mild pixel movement does not affect reality, but the image still signifies plantlings. It would not have been possible to employ such an extensive network without data augmentation since it would have suffered from significant overfitting. For data augmentation, two operations (horizontal flipping and vertical flipping) are performed at angles of 45 and 180 degrees for all the 51 images. This resulted formation of 4 replicas for each image. The mathematical formula for image augmentation using flipping method is given in Eq (4).

$$A = \begin{pmatrix} Cos\ \theta & -Sin\ \theta \\ Sin\ \theta & Cos\ \theta \end{pmatrix} \tag{4}$$

**3.2.3. Dropout.** A neuron leaves the network with a probability of 0.5 to 0.25 in dropout. When a neuron goes, it has no influence on either forward or backward propagation. As a result, each input is directed through new network architecture [61]. The learned weights

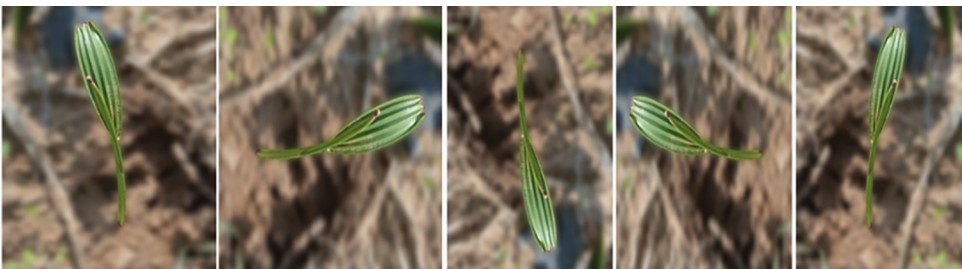

**Fig 5. Data augmentation process.**

parameters are more resilient and do not quickly get overfitted. During test results, there is no attrition, and the entire network has been used. However, the outcome is inflated by a factor of 0.25 to compensate for neurons that were neglected while retraining.

## 4 Experimentation and results

### 4.1 Dataset collection

To evaluate the proposed method, a new data set of palm tree seedlings was created. Our data set comprises pictures of palm tree seedlings that were captured with UAV technology. All the image data we used in our experiment came from the nursery garden of V.M. Arockianatha-puram in Tirunelveli City, Tamil Nadu, India (8˚41'51.4"N 77˚46'54.9"E), where it was taken in December 2022. The average temperature for the year is 24.5 degrees Celsius. The nursery is 1,500 square meters and is lit by natural light. To keep the variety of palm tree seedling images, it is essential to consider a range of vertical heights when gathering information about these images. Also, pictures are taken both in the morning and in the afternoon. The pictures came from the same group of palm tree seedlings that had been growing for ten days and were taken in different kinds of light. After the images have been gathered, they are shrunk to 128x128 pixels. Fig 6 shows the specific images of the seedlings that were taken.

The dataset is created with 51 images initially. For the created dataset, data augmentation results in the creation of 4 replicas for each image. The replicas are made with an angular shift of 45 and 180 degrees through horizontal and vertical flipping. The images are subjected to data augmentation, resulting in the creation of 255 images in the dataset. The training vs testing ratio is realized as 80:20. In the dataset, 204 were put into the training set, and 51 were put into the testing set. The utilized dataset is publicly available in the Kaggle repository at https://www.kaggle.com/datasets/rajmohan89/palm-tree-plantlings-health-prediction.

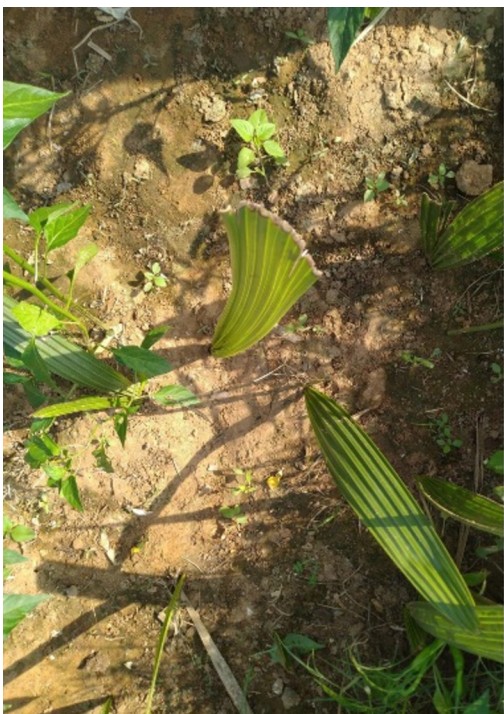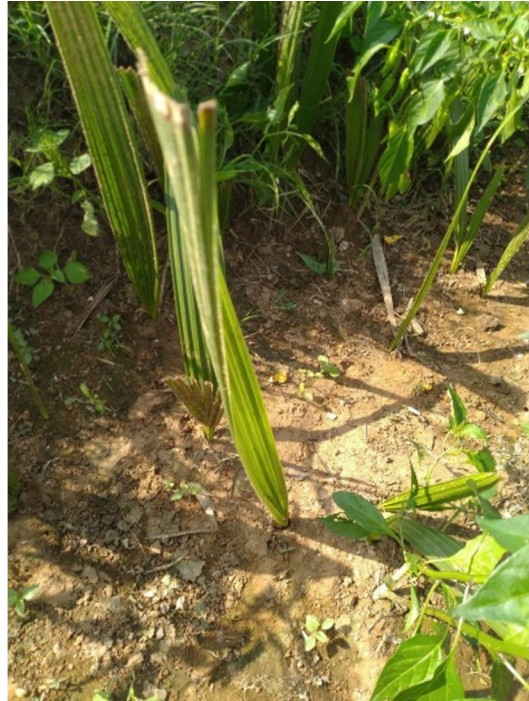

**Fig 6. Sample palm tree plantlings.**

Every oil palm seedling has grown in different temperatures and humidity levels. And the images are generated at different places where palm plantlings are germinated. Every plant has some desirable temperature to grow and survive. The healthy palm seedling requires well-drained topsoil and thrives in strongly alkaline conditions. The successful growth of plantlings is satisfied only under the minimum temperature range between 22˚C to 24˚C (71.6˚F-75.4˚F), the maximum temperature range between 20˚C to 33˚C (68˚F -91.4˚F), and the optimal temperature ranges from 30˚C–32˚C (86˚F–89.6˚F). Ensure that the topsoil is consistently saturated and when the upper region of soil is dry, water is supplied to the palms. Moreover, screened sunshine is preferable for palm seed germination. A humidity level of approximately 60–80 percent is desirable. These are the factors that are included in this research.

The plantlings are categorized into two classes depending on their growth and development. The seedling is said to be poor if any black spot is formed in the tiny leaf due to lack of nutrients if the temperatures and humidity levels are not optimal, and the leaf looks dried yellowish. Fig 7 represents successful growth, and Fig 8 indicates poor growth (lack of nutrients due to the unsuitable temperature). The IoT model is utilized to identify the ground truth of the palm tree plantling health conditions. The UAV technology is used for capturing images of the palm tree plantlings. The IoT kit (Fig 9) is a grove sensor (DHT11) in which the ground is connected to the Raspberry Pi. This model was considered as it is based on long-term stability and low power consumption. Connect this sensor to the respective ports using a cable that connects both the grove sensor and the Raspberry pi. Moreover, the raspberry pi is connected to the electrical supply using a small USB cable. This kit is utilized to measure the temperature and the humidity level of the plantlings where they grow. The normal temperature range for oil palm trees is 20˚C -33˚C. The humidity varies from 60-to 80 percent. And so, the measured

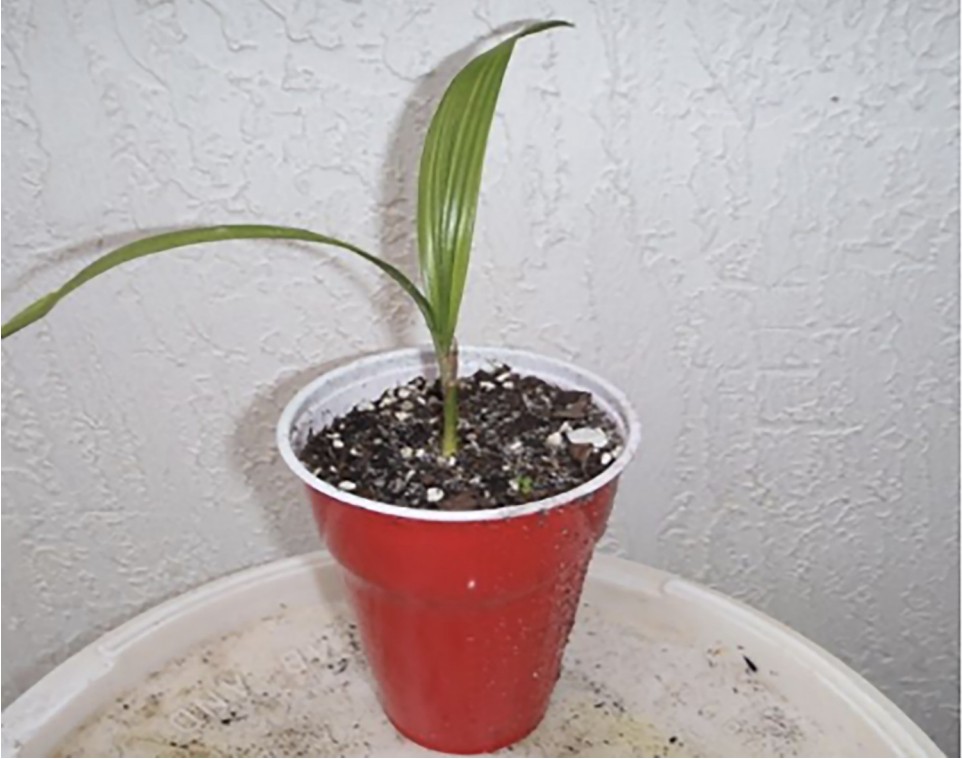

**Fig 7. Healthy seedling growth.**

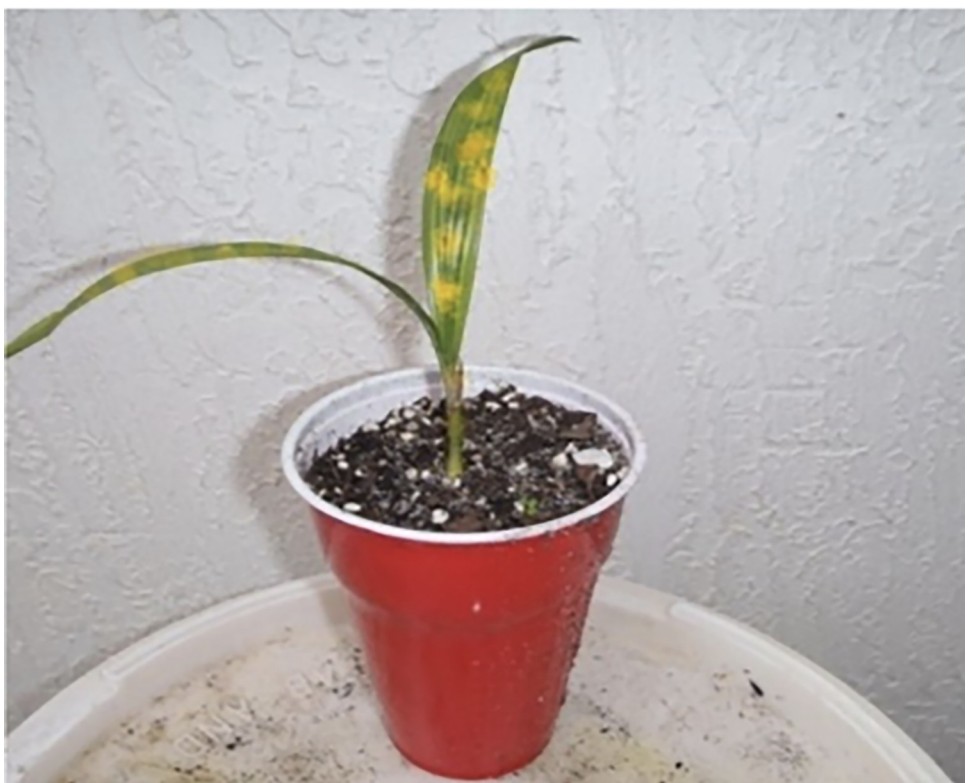

**Fig 8. Poor seedling growth.**

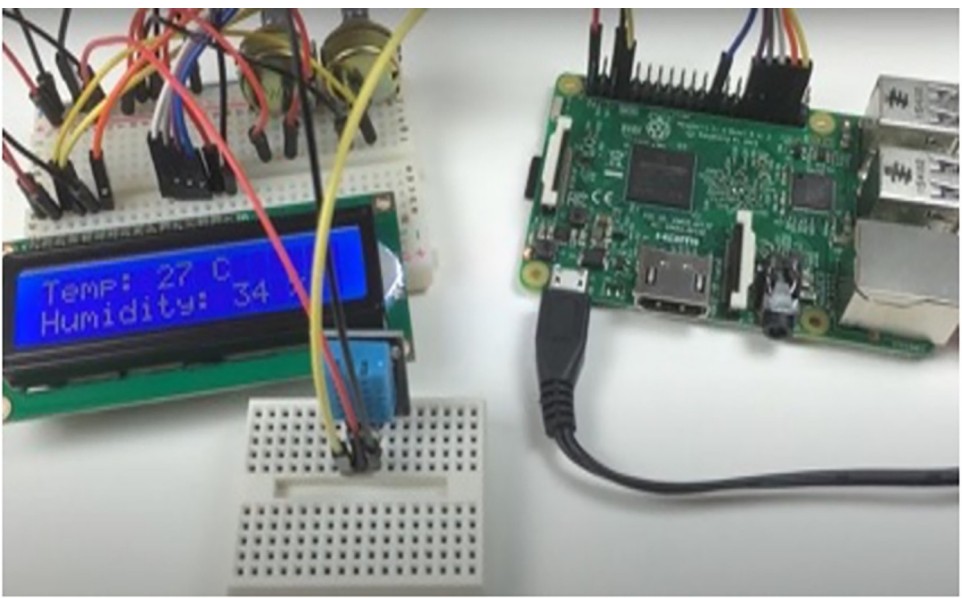

**Fig 9. IoT monitoring module.**

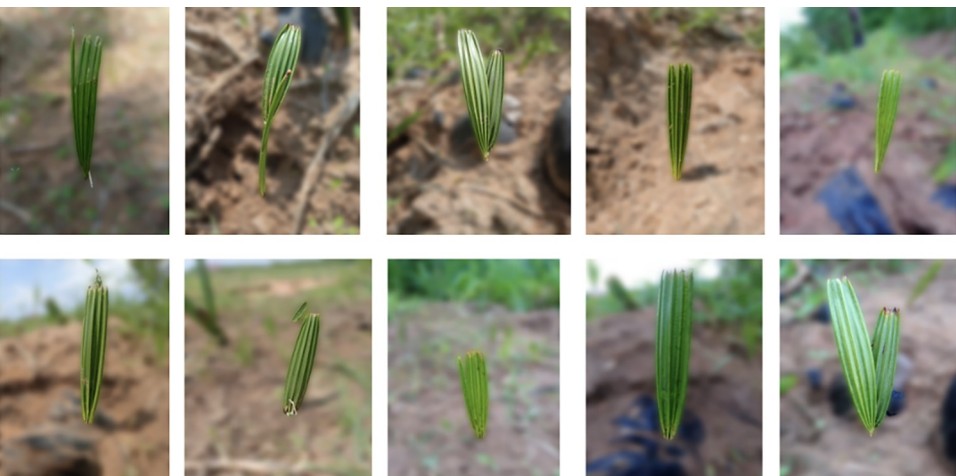

**Fig 10. Preprocessed palm plantlings dataset images.**

levels satisfy the original and are maintained at the proper temperature, then the plantlings will have enough nutrients to grow successfully. If temperature and humidity levels are not satisfied, it will make the seedling discolored and suffer from nutrient deficiency. They are detecting the humidity and temperature level of the captured plantlings to find their growth prediction. Fig 10 visualizes the preprocessed images of palm tree plantlings.

- **STEP 1: DATA VISUALISATION:** Although our training dataset encompasses a significant collection of photos, we would commence by graphing the images that correspond through the most categories. Successful growth refers to plantlings that have developed adequately under certain temperatures and humidity. The plantlings with a black dot or any yellow pigment due to the nutrient less are spotted and classified as having poor growth.

- **STEP 2: DATA AUGMENTATION:** More images were uploaded to the collection for data augmentation, and each picture was flipped as we progressed through this cycle. We had some additional images after using the data augmentation approach, with some images falling into the 'Successful growth' category and some may be falling into the 'Failure' category under the 'poor growth' label.

- **STEP 3: MODEL VERIFICATION:** With 250 epochs, this phase strives to fit the pictures from the training and validation datasets into the Framework. The use of the cross-validation methodology eliminates data fitting problems. For epoch = 250, got a result with loss: 0.1149, accuracy: 0.9823, val_loss: 0.5312, val_accuracy: 0.9610. Our model takes 15 sec to execute one epoch. Table 2 explains the training and testing values obtained for loss and accuracy predictions in palm tree seedling prognosis.

- **STEP 4: DATA CATEGORIZATION:** In this case, n symbolizes the n-th instance, while $k \in \{0, 1 \ldots K-1\}$ designates one certain class. Optimizing log-likelihood is just the same as lowering categorical cross-entropy damage. It is expressed in Eq 5.

**Table 2. Accuracy and loss.**

| | |
|---|---|
| TEST LOSS: 0.156 | TRAIN LOSS: 0.1149 |
| TEST ACCURACY: 0.9610 | TRAIN ACCURACY: 0.9823 |

$$N = -log \log p\left(\frac{y^n}{x(t)n}\right) = -\sum_{k=0}^{K} n \, y^n \log(y^n) k \tag{5}$$

*where* $n = k = 0 \; or \; 1$

Here log $(y^n)$ specifies an indicator function for the tag, and n = 2 in our scenario. The net loss for our dataset is expressed by: $\sum_{k=0}^{K} n \, y^n$. Assigning a cost to all the classifications compensates for both the categorical cross-entropy reduction. This accentuates the minority class, especially for unlabeled data. As an outcome, the design tries to reduce misperception of this class. When the image returns with a classifier value of 0, the growth goes well. If the image is returned with the value 1 for the classifier, it is marked as having poor growth, which means that the attempt to grow failed.

## 4.2 Performance analysis

The entire approach of training and validation of the seedling growth prediction model reported in this study was carried out on a single workstation. The Neural network is trained using a graphics processing unit (GPU). A workstation with an i7 GPU from the 11th generation and 16 GB of RAM was used to test the proposed methods. The implementation was written in Python 3.10.5 using Tensor and Keras libraries. This implementation realized 250 epochs, ReLU optimizer, and a learning rate of 0.0001. The proposed system produces the outcome shown in Fig 11. The results indicate that the seedling growth is flourishing. The findings reported in this part are from testing with the entire data set, which comprises both original and augmented images.

A simpler architecture will increase prediction accuracy (as shown in Fig 12) and may be more efficient in capturing the data structure than many of the larger versions employed in the previous study [62]. Epoch sizes can enhance the accuracy up to a certain point, after which the simulation moves into overfitting. Possessing a shallow one will lead to underfitting as well.

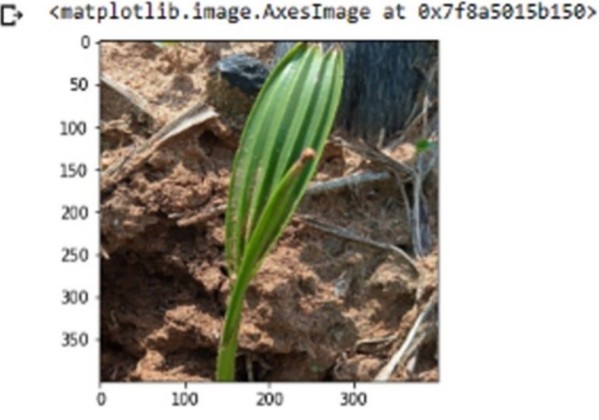

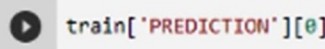

**Fig 11. Growth prediction.**

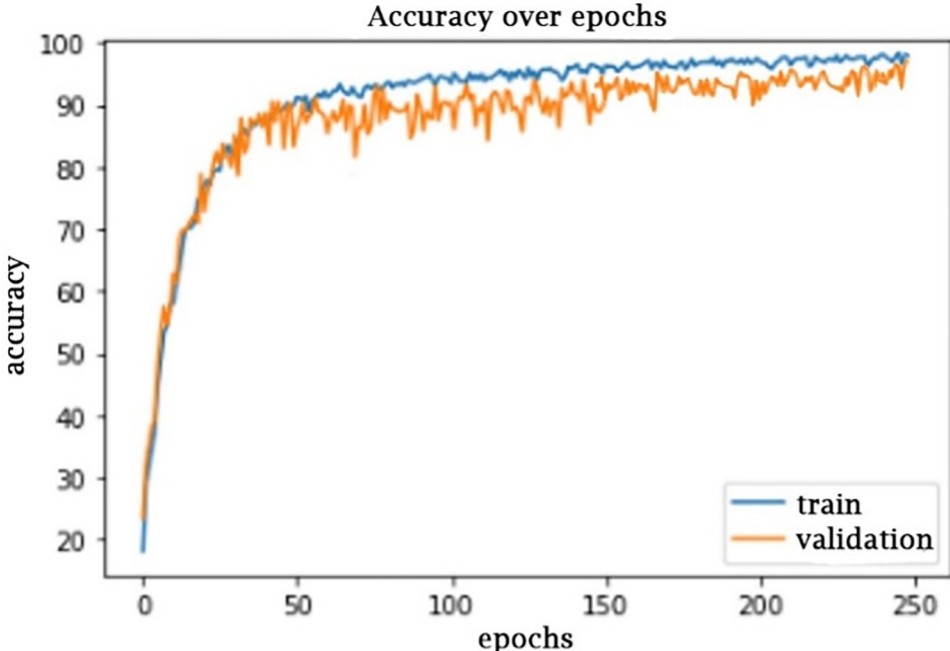

**Fig 12. Accuracy with 250 epochs.**

As a result, the proposed CNN architecture seems to be simpler and provides better performance, which is shown in Fig 12. MAE is an abbreviation for the Average of All Intrinsic Losses. The framework was tested to establish the model's loss and accuracy values. The model's Mean Absolute Error (Fig 13) is really the arithmetic mean values of each standard error above all repetitions of the testing data.

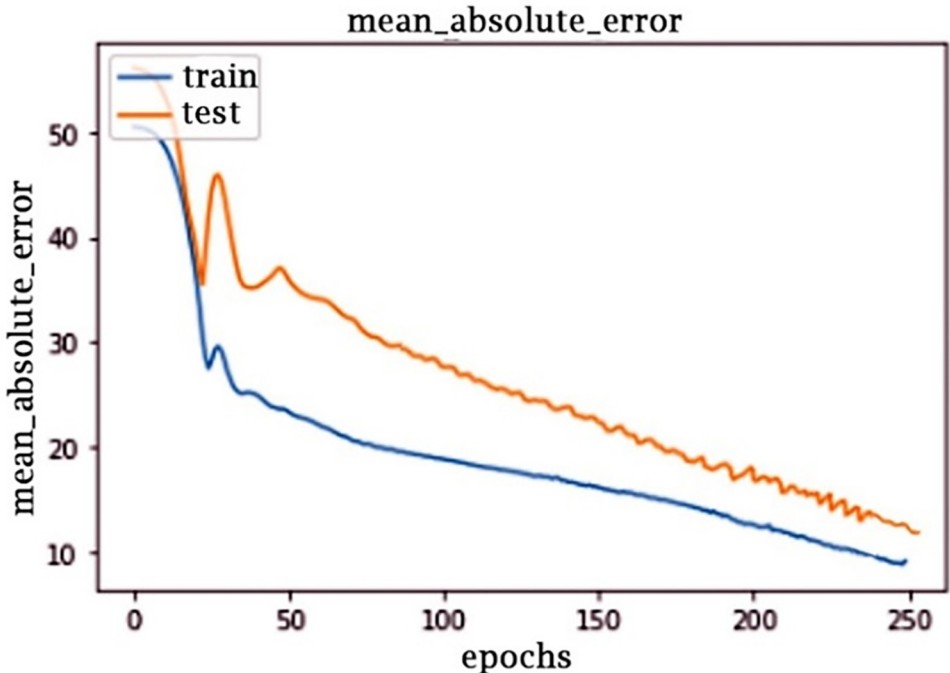

**Fig 13. Mean absolute error over 250 epochs.**

**Table 3. Performance of GL-CNN vs ground truth.**

| Class | Ground Truth | GL-CNN | Accuracy | Precision | Recall | F1 Score |
|---|---|---|---|---|---|---|
| Healthy | 231 | 222 | 96.10 | 0.967 | 0.924 | 0.987 |
| Unhealthy | 24 | 23 | 95.83 | 0.982 | 0.978 | 0.978 |

## 5 Discussions

The trained model was evaluated on each class autonomously. Every palm tree sampling image from the validation data was inspected. The statistics are provided to showcase how many successful versus poor development seedling pictures are reliably predicted from the overall of each class. Using the Soft max activation function formula, the CNN output layer will estimate the probability of each class. GL-CNN generates the $i^{-th}$ neuron with the peak power probabilities score projection. Table 3 and Fig 14 exhibit the predictive accuracy of the phase for each category. Experiments indicated that the proposed implemented system monitors growth and consumes less inference speed and amount of memory than existing methods. We examined the correctness of the prior techniques to the suggested one to further illustrate the new model's success. The comparison is shown in the Fig 15. It outperforms the other models and proves that the error rate is low compared to the traditional methods. Here, the value 0 denotes healthy growth and value 1 denotes poor growth. Our method obtained an accuracy of 96.10 in predicting healthy growth and 95.83 in predicting poor growth.

After the training phase, we used a test set of 51 images of palm tree seedlings to test how well the proposed model worked and how well it worked in general. This paper uses the accuracy rate A, the precision rate P, the recall rate R, and the F1 score to measure how well our

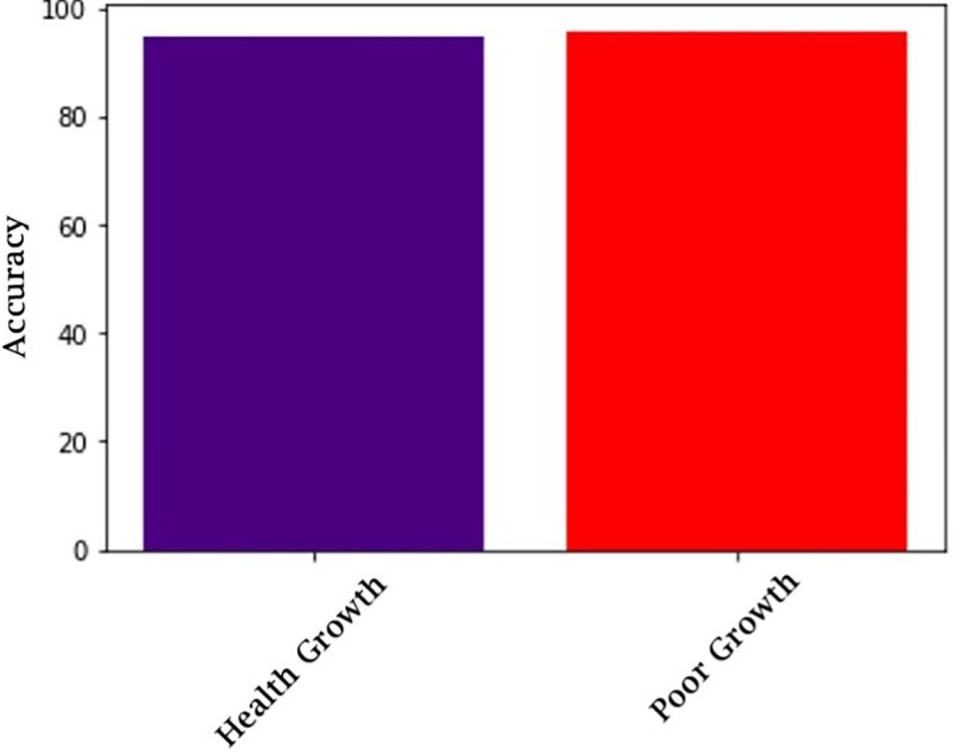

**Fig 14. Prediction accuracy.**

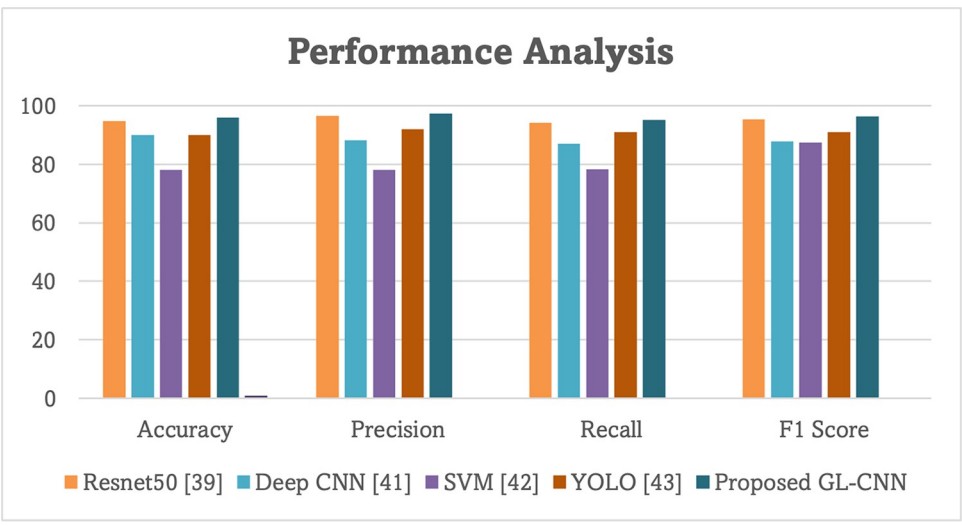

**Fig 15. Prediction performance analysis.**

model works. In Eqs 9, 10, and 11, the definitions of P, R, and F1 are provided.

$$P = \frac{TP}{TP + FP} \tag{6}$$

$$R = \frac{TP}{TP + FN} \tag{7}$$

$$F1 = \frac{TP}{TP + \frac{1}{2}(FP + FN)} \tag{8}$$

True Positives are denoted by TP, False Positives by FP, and False Negatives by FN. Furthermore, TP denotes the appropriate detection box, FP denotes the false detection box, and FN denotes the missing detection box. The detection box is a simplified representation of plantlings found in a specific image.

We compared our model to others, such as Resnet50 [39], Deep CNN [41], SVM [42], and YOLO [43] to determine how accurately each could predict the growth of plant seedlings. The performance of each model on the same test set is displayed in Table 4 and Fig 15, respectively, after training several distinct detection frameworks until convergence.

Table 4 and Fig 15 depict the results of the suggested strategy, compared against five other recent approaches. The existing methods attained a maximum of 94.87 accuracy value, 97.08 precision value, 95.21 recall value and 96.28 F1 value. Among the existing approaches,

**Table 4. Performance of GL-CNN vs existing models.**

| Framework | Accuracy | Precision | Recall | F1 Score |
|---|---|---|---|---|
| Resnet50 [39] | 94.87 | 96.54 | 94.09 | 95.30 |
| Deep CNN [41] | 90.08 | 88.15 | 87.11 | 87.75 |
| SVM [42] | 78.12 | 78.10 | 78.25 | 87.38 |
| YOLO [43] | 90 | 92 | 91 | 91.02 |
| Proposed GL-CNN | 95.96 | 97.45 | 95.10 | 96.36 |

Resnet50 predicted the growth of palm tree plantlings more efficiently than other frameworks. In comparison with existing methods, the proposed technique achieved 95.96 accuracy value, 97.45 precision value, 95.10 recall value and 96.36 F1 score. GL-CNN is compared to other CNNs, found that it outperforms the traditional methods which were implemented in previous research.

## 6 Conclusion

Palms tree are one of the most important economic plants and with the developing technology, their indigenous growth can be monitored. Plant growth predicting in a real-world environment requires high accuracy and robustness. Currently, UAVs are widely used in agriculture as they can capture images with high spatial resolution. IoT technology has enables smart monitoring of palm tree plantlings growth and their health prediction. Combining the advantage of UAV, IoT and CNN model, we proposed a novel GL-CNN model for growth prediction in palm tree plantlings. The GL-CNN has improved the prediction performance by inducing normalizing layer, the global average pooling layer, and the depth of the pooling layer filters. UAV was utilized for capturing high quality spectral images and IoT module is used for obtaining the ground truth information of palm tree plantlings health. Using these two technologies, a new dataset of 255 images of palm tree plantlings was collected and made freely public for other researchers. The evaluation results, using this dataset, showed that the GL-CNN model performs with an average accuracy of 95.96% for predicting the palm plantlings growth. Comparing with related work including the original CNN, these results showed to be the best.

The limitation of the proposed model relies on not considering the endogenous metabolites within the palm tree plant, for growth prediction. Moreover, the IoT module is not supported with electrochemical and optical sensors for analyzing the surface metabolites. Realizing these parameters and sensors in the proposed model would enhance the growth prediction process. In the future, it will be fascinating to expand the technique to a variety of agriculturally important species to develop a library of trained networks. Also, we employed normal RGB photos and not evaluated the plant seedling growth during the night. This may lead to overlooked occurrences that would have shifted our evaluation of the plantlings' phases of growth. Machine vision cameras, which are inexpensive, could be utilized to gain accessibility to night events. Additionally, Bayesian techniques, such as polynomial interpolation, could be used to determine the amount of time required to obtain potentially missing data.

## Author Contributions

**Conceptualization:** T. Ananth Kumar, Sunday Adeola Ajagbe.

**Data curation:** Sunday Adeola Ajagbe.

**Investigation:** Sunday Adeola Ajagbe, Tarek Gaber.

**Methodology:** T. Ananth Kumar, Sunday Adeola Ajagbe.

**Project administration:** R. Rajmohan.

**Resources:** R. Rajmohan, Fatma Masmoudi.

**Software:** R. Rajmohan.

**Supervision:** T. Ananth Kumar, Tarek Gaber, Xiao-Jun Zeng.

**Validation:** Tarek Gaber, Xiao-Jun Zeng.

**Visualization:** R. Rajmohan.

**Writing – original draft:** Sunday Adeola Ajagbe, Xiao-Jun Zeng.

**Writing – review & editing:** Tarek Gaber, Fatma Masmoudi.

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
