## [Decision Letter · Decision Letter 0]

20 Feb 2023

PONE-D-23-01139CNN-SLNet: A novel CNN architecture for growth prediction of palm tree plantlingsPLOS ONE

Dear Dr. Gaber,

Thank you for submitting your manuscript to PLOS ONE. After careful consideration, we feel that it has merit but does not fully meet PLOS ONE’s publication criteria as it currently stands. Therefore, we invite you to submit a revised version of the manuscript that addresses the points raised during the review process.

We look forward to receiving your revised manuscript.

Kind regards,

Muhammad Attique Khan

Academic Editor

PLOS ONE

Journal Requirements:

2. Please note that PLOS ONE has specific guidelines on code sharing for submissions in which author-generated code underpins the findings in the manuscript. In these cases, all author-generated code must be made available without restrictions upon publication of the work. 

Please review our guidelines at https://journals.plos.org/plosone/s/materials-and-software-sharing#loc-sharing-code and ensure that your code is shared in a way that follows best practice and facilitates reproducibility and reuse.

4. Please ensure that you refer to Figures 1, 2, 4, 7, 8, and 12 in your text as, if accepted, production will need this reference to link the reader to the figure.

Reviewers' comments:

Reviewer's Responses to Questions

**Comments to the Author**

1. Is the manuscript technically sound, and do the data support the conclusions?

Reviewer #1: No

Reviewer #2: Yes

2. Has the statistical analysis been performed appropriately and rigorously? 

Reviewer #1: I Don't Know

Reviewer #2: Yes

3. Have the authors made all data underlying the findings in their manuscript fully available?

Reviewer #1: No

Reviewer #2: Yes

4. Is the manuscript presented in an intelligible fashion and written in standard English?

Reviewer #1: Yes

Reviewer #2: Yes

5. Review Comments to the Author

Reviewer #1: Authors should address the following major revision.

1) The tree has the potential to reduce both the temperature of the atmosphere and the level of pollutants produced by industrial activity. The date palm tree's symmetrical design has introduced a new aspect to its consequences for future environmental betterment. - add the justification of this sentence through proper reference.

2) Add a clear problem statement of this work under the introduction section. Also, improve the bullets of significant contributions.

3) “In this study, we design a novel CNN-SLNet architecture for efficient monitoring and prediction of palm tree seedling health using UAV and IoT technology”. -Explain, how authors used the IoT technology?

4) At the end of the Related Work section, add some gaps and cutting edges of the recent techniques.

5) How many image are set after the augmentation process? What is the training and testing ratio?

6) How many operations are performed for the data augmentation? It is better to add mathematical formulas.

7) Add Experimental setup under the performance metrics section.

8) Consider adding recent works like those below

• Abbas, Shafaq, et al. "Crops Leaf Diseases Recognition: A Framework of Optimum Deep Learning Features." (2023).

• Khan, Muhammad Attique, et al. "Cucumber leaf diseases recognition using multi-level deep entropy-ELM feature selection." Applied Sciences 12.2 (2022): 593.

• Khan, Muhammad Attique, et al. "Fruits diseases classification: exploiting a hierarchical framework for deep features fusion and selection." Multimedia Tools and Applications 79 (2020): 25763-25783.

9) Figure 1, 11, 12, 13 and 14. Present these Figures in a better quality.

10) Improve the conclusion and Add dark sides of the proposed model in the conclusion.

Reviewer #2: The performance of the existing CNN-based models for Counting and identifying plantlings could be further improved. To achieve this, a novel CNN architecture (CNN-SLNet) has been proposed and used for building a new IoT effective monitoring system for the prognostication of oil palm tree seedling. The proposed model is trained to predict the successful and poor seedling growth for a given set of palm tree seedling images. The proposed CNNSLNet architecture is novel in terms of defined con-volution layers and the gap layer designed for output classification.

Why the tree has the potential to reduce both the temperature of the atmosphere and the level of pollutants produced by industrial activity? You need to validate “automated monitoring and growth interpretation of palm tree seedlings gives farmers a new way to manage their resources that is based on technology instead of the old way they did it in the past”.

Why the potential to comprehend complex data is a substantial advantage of deep learning? How deep learning has been implemented to crop cultivation to cut production costs and hence increase agricultural productivity?

Section 2, pls discuss related papers, fruit leaf diseases classification: a hierarchical deep learning framework, intelligent tracking of mechanically thrown objects by industrial catching robot for automated in-plant logistics 4.0, fruit category classification by fractional Fourier entropy with rotation angle vector grid and stacked sparse autoencoder.

How recent advancements in software platform, wireless sensors, and computer vision may allow for significant time and expense reductions in plant seedling monitoring? What is the motivation to design a novel CNN-SLNet architecture for efficient monitoring and prediction of palm tree seedling health using UAV and IoT technology?

Why this makes it challenging to keep a close eye on the seeds and young plants. Restoration professionals require technical approaches that can provide them with highresolution, rapid, and scalable plant-based monitoring systems?

Description of CNN is unclear. You can refer to two-stage intelligent darknet-squeezenet architecture-based framework for multiclass rice grain variety identification, a five-layer deep convolutional neural network with stochastic pooling for chest CT-based covid-19 diagnosis.

Why Baye’s optimizing tests also confirmed that ReLU and dropout have collaboration, which implies that using them together is optimal?

Virtue of global average polling is not clear. Please check “Global average pooling has virtue over fully connected layers (FC) layers in that it promotes correspondences amongst feature maps and subcategories, making it more ideal for the convolution layout. As a corollary, the extracted features may be simply under-stood as category optimism maps”

6. PLOS authors have the option to publish the peer review history of their article (what does this mean?). If published, this will include your full peer review and any attached files.

Reviewer #1: No

Reviewer #2: No

---

## [Author Response · Author response to Decision Letter 0]

30 Apr 2023

Responding to the comments raised by the editor and the reviewers, we are uploading three files: 

(a) our point-by-point response to the comments (response to reviewers), 

(b) an updated manuscript with yellow highlighting indicating changes, and 

(c) a clean updated manuscript without highlights

---

## [Decision Letter · Decision Letter 1]

31 Jul 2023

A NOVEL CNN GAP LAYER FOR GROWTH PREDICTION OF PALM TREE PLANTLINGS

PONE-D-23-01139R1

Dear Dr. Gaber,

We’re pleased to inform you that your manuscript has been judged scientifically suitable for publication and will be formally accepted for publication once it meets all outstanding technical requirements.

Kind regards,

Mohamed Hammad, Ph.D.

Academic Editor

PLOS ONE

Additional Editor Comments (optional):

Reviewers' comments:

Reviewer's Responses to Questions

**Comments to the Author**

1. If the authors have adequately addressed your comments raised in a previous round of review and you feel that this manuscript is now acceptable for publication, you may indicate that here to bypass the “Comments to the Author” section, enter your conflict of interest statement in the “Confidential to Editor” section, and submit your "Accept" recommendation.

Reviewer #1: All comments have been addressed

2. Is the manuscript technically sound, and do the data support the conclusions?

Reviewer #1: Partly

3. Has the statistical analysis been performed appropriately and rigorously? 

Reviewer #1: I Don't Know

4. Have the authors made all data underlying the findings in their manuscript fully available?

Reviewer #1: Yes

5. Is the manuscript presented in an intelligible fashion and written in standard English?

Reviewer #1: Yes

6. Review Comments to the Author

Reviewer #1: Your commitment to enhancing the work and ensuring its quality is truly commendable. The revisions you've made have significantly improved the overall clarity and coherence of the content, making it even more engaging and insightful.

7. PLOS authors have the option to publish the peer review history of their article (what does this mean?). If published, this will include your full peer review and any attached files.

Reviewer #1: No

---

## [Editor Report · Acceptance letter]

3 Aug 2023

PONE-D-23-01139R1 

A NOVEL CNN GAP LAYER FOR GROWTH PREDICTION OF PALM TREE PLANTLINGS 

Dear Dr. Gaber:

I'm pleased to inform you that your manuscript has been deemed suitable for publication in PLOS ONE. Congratulations! Your manuscript is now with our production department. 

Kind regards, 

on behalf of

Dr. Mohamed Hammad 

Academic Editor

PLOS ONE